# O-GlcNAcylation Affects the Pathway Choice of DNA Double-Strand Break Repair

**DOI:** 10.3390/ijms22115715

**Published:** 2021-05-27

**Authors:** Sera Averbek, Burkhard Jakob, Marco Durante, Nicole B. Averbeck

**Affiliations:** 1Department of Biophysics, GSI Helmholtzzentrum für Schwerionenforschung GmbH, 64291 Darmstadt, Germany; s.averbek@gsi.de (S.A.); b.jakob@gsi.de (B.J.); m.durante@gsi.de (M.D.); 2Department of Biology, Technische Universität Darmstadt, 64287 Darmstadt, Germany; 3Department of Physics, Technische Universität Darmstadt, 64287 Darmstadt, Germany

**Keywords:** O-GlcNAcylation, DNA-DSB repair, chromatin remodeling, high LET, particle irradiation, ionizing radiation

## Abstract

Exposing cells to DNA damaging agents, such as ionizing radiation (IR) or cytotoxic chemicals, can cause DNA double-strand breaks (DSBs), which are crucial to repair to maintain genetic integrity. O-linked β-N-acetylglucosaminylation (O-GlcNAcylation) is a post-translational modification (PTM), which has been reported to be involved in the DNA damage response (DDR) and chromatin remodeling. Here, we investigated the impact of O-GlcNAcylation on the DDR, DSB repair and chromatin status in more detail. We also applied charged particle irradiation to analyze differences of O-GlcNAcylation and its impact on DSB repair in respect of spatial dose deposition and radiation quality. Various techniques were used, such as the γH2AX foci assay, live cell microscopy and Fluorescence Lifetime Microscopy (FLIM) to detect DSB rejoining, protein accumulation and chromatin states after treating the cells with O-GlcNAc transferase (OGT) or O-GlcNAcase (OGA) inhibitors. We confirmed that O-GlcNAcylation of MDC1 is increased upon irradiation and identified additional repair factors related to Homologous Recombination (HR), CtIP and BRCA1, which were increasingly O-GlcNAcyated upon irradiation. This is consistent with our findings that the function of HR is affected by OGT inhibition. Besides, we found that OGT and OGA activity modulate chromatin compaction states, providing a potential additional level of DNA-repair regulation.

## 1. Introduction

DNA is constantly subjected to a variety of DNA-damaging agents both endogenously, such as hydrolysis, oxidation and alkylation, or exogenously such as ionizing radiation (IR), ultraviolet radiation (UV) and cytotoxic chemicals, leading to single-strand breaks, base damages or even DSBs [1,2]. Amongst them DSBs are considered the most dangerous type of DNA lesions [3] as their misrepair or failure of repair paves the way for detrimental consequences such as genome instability and mutagenesis, which can ultimately lead to cancer [4]. The density and complexity of DNA damage caused by ionizing radiation is linked to radiation quality, which can be classified using linear energy transfer (LET). High LET radiation like α particles, carbon or other heavy-ions-induced DNA damage is more clustered, and hence more difficult to repair than damage inflicted by low-LET radiation (X or γ-rays) [2,5].

After irradiation, one of the earliest steps in DDR is the recruitment of the MRN (MRE11/RAD50/NBS1) complex to DSBs that start to accumulate after a few seconds and lead to radiation induced foci (RIF) in living cells after sparsely or densely ionizing irradiation [6,7]. One of the key roles of the MRN complex is the activation and boosting of the performance of ATM (ataxia-telangiectasia mutated), a member of PI-3-like kinases. ATM is a central player in DDR signaling and, besides other substrates, activates ATM phosphorylates histone H2A variant H2AX generating γH2AX [8]. γH2AX is one of the most prominent markers of chromatin containing DSB sites and is frequently used as a surrogate marker of DSBs in immunocytochemistry [9]. Thus, by interaction with ATM and MDC1, the MRN complex participates in recognition, stabilizing and downstream signaling of DSBs to control cell cycle, DNA repair pathway choice and ultimately survival [10,11,12].

In mammalian cells, the repair of DNA DSBs is mainly governed by two primary pathways: nonhomologous end joining (NHEJ), and homologous recombination (HR). NHEJ is the most common and prominent DSB repair pathway due to its activation throughout the cell cycle, whereas HR activity is mostly limited to the (late) S and G2 phase of the cell cycle, since HR relies mainly on the replicated sister chromatid as a template. As HR-mediated DSB repair is based on the homologous sequence to restore genomic information, it is considered to be largely error-free [13]. To conduct necessary strand invasion and D-loop formation, substantial resection of the double-stranded DNA at the break sites is required [13]. The MRN complex interacts with CtIP and, in cooperation with Exo1 and DNA2, performs resection to generate 3′ single-stranded DNA (ssDNA) overhangs at the DSB ends [14].

O-linked β-N-acetylglucosamine (O-GlcNAc) is a highly dynamic and reversible post-translational modification of nuclear, mitochondrial and cytoplasmic proteins that plays a crucial role in regulating numerous biological processes via activity, localization, or stability of substrate proteins. O-GlcNAcylation is dynamically catalyzed by O-GlcNAc transferase (OGT), which uses UDP-GlcNAc as a substrate to attach GlcNAc moieties to protein Ser/Thr residues, whereas O-GlcNAcase (OGA) catalyzes the removal of O-GlcNAc from O-GlcNAc modified proteins [15,16]. The dynamics of O-GlcNAcylation is tightly regulated and influenced by the environmental availability of glucose, glutamine and glucosamine [17].

Besides direct regulation of protein activities, O-GlcNAcylation can compete with phosphorylation at the same Ser/Thr sites [16], yielding an additional level of interference with the DNA-damage response, which is known to be widely regulated via phosphorylation [18]. Indeed, earlier work has affirmed this link [19,20,21]. Here, we address in more detail how O-GlcNAcylation influences DNA DSB repair with dependence on radiation quality, the cell cycle and utilized repair pathways. In addition, we identify factors of the DDR modulated or regulated by O-GlcNAcylation.

According to recent reports, O-GlcNAcylation not only modifies repair factor activity but regulates chromatin compaction by interfering with other posttranslational modifications of histones [22,23,24] and indirectly by its complex relationship with the PcG (Polycomb-group) proteins and the TET (Ten-Eleven Transcription) family proteins [23,24,25]. The role of O-GlcNAc on chromatin architecture in the context of the DDR has not been studied yet. However, since the chromatin status is crucial for DSB-repair processes [26], O-GlcNAcylation might impact on DSB repair that way as well. To this end, we made use of a recently established method based on FLIM [27,28] to identify effects of O-GlcNAcylation on the chromatin compaction status after treating the cells with OGT or OGA inhibitors.

Our data suggest that O-GlcNAcylation is important for DSB repair in several ways. It is important for the chromatin status, functional HR and the DSB retention of NBS1, one crucial factor of the MRN complex at the DSB break site.

## 2. Results

### 2.1. O-GlcNAcylation Is Differently Upregulated after Low and High LET Irradiation

In order to find out whether radiation induced DSBs lead to a local modification of O-GlcNAcylated proteins, we studied the colocalization of O-GlcNAc and the DSB marker 53BP1 shortly (20 min) after X-ray irradiation via immunofluorescence microscopy in MCF-7 cells. After X-ray irradiation, we did not observe DSB site specific O-GlcNAcylation (Figure 1a). However, the nuclear O-GlcNAc level was clearly increased after 10 Gy of X-ray irradiation (Figure 1a) suggesting a more global radiation induced O-GlcNAcylation. A similar nuclear wide enrichment of O-GlcNAcylation was observed in HeLa cells (data not shown). Interestingly, upon iron-ion irradiation, DSBs sites were clearly decorated with O-GlcNAc 20 min after irradiation (Figure 1b). Intensity profiles along the yellow lines in Figure 1b, confirmed the colocalization of O-GlcNAc and 53BP1 foci (Figure 1c). Quantitative analysis of colocalization revealed that about 80% of 53BP1 foci colocalized with areas of increased O-GlcNAc levels (Figure 1d). In addition, global, nuclear wide O-GlcNAcylation levels were also increased after iron-ion irradiation (Figure 1e). Overall, our results show that O-GlcNAcylation is not only increased in response to DNA damge but, at least after high LET irradiation, occurs increasingly at DSBs sites.

### 2.2. O-GlcNAcylation Impacts DSB Repair after X-ray or Heavy Ion Irradiation

#### 2.2.1. O-GlcNAcylation Is Important for the Repair of X-ray Induced DSBs in S/G2 Phase of the Cell Cycle

Earlier work has shown that disturbing O-GlcNAcylation influences DSB repair [20,25]. Here, we studied the influence of OGT and OGA inhibition, respectively, on DSB rejoining in a cell-cycle dependent manner. As the activity of different DSB-repair pathways partially depends on the cell-cycle stage (e.g., HR is mainly active in S and G2 phase) this may give a hint to whether a particular DSB repair pathway is influenced by O-GlcNAcylation. To study the influence of O-GlcNAcylation on repair of X-ray-induced DSBs in a cell-cycle dependent manner, γH2AX foci assays were performed in HeLa cells with dependence on functional OGT or OGA (Figure 2a–c). The cell-cycle phases were determined by coimmunostaining centromere protein F (CENP-F), which is expressed in S, G2 and M phase, but hardly in G1 phase [29], and counterstaining DNA with DAPI, which indicates M-phase cells. OGT was inhibited with ST060266 and OGA with PUGNAc. The former inhibitor acts as a suppressant of O-GlcNAcylation while the latter elevates protein O-GlcNAcylation (Appendix A).

At first, we assured that the OGT and OGA inhibition did not influence the number of radiation-induced DSBs, as this would distort the analysis of the rejoining data analysed 24 h after irradiation. Hence, we quantified γH2AX foci 15 min after X-ray irradiation (1 Gy), a time point when radiation-induced DSBs had become clearly visible by γH2AX staining. Neither of the inhibitors influenced the number of induced DSBs (Figure 2b). Whereas repair in G1 phase was largely unaffected (Figure 2c), quantifying the γH2AX foci numbers 24 h after irradiation (6 Gy) showed a significant modulation of DSB rejoining in S/G2 phase cells by both inhibitors. Inhibiting O-GlcNAcylation (OGTi) impaired DSB repair substantially, while inhibiting the removal of O-GlcNAcylation (OGAi) improved it (Figure 2c).

Previously, Chen and Yu reported that spreading of H2AX phosphorylation is limited by OGT-dependent O-GlcNAcylation of H2AX in response to laser microirradiation [21]. As the number of initially detected γH2AX foci was not changed by O-GlcNAcylation (Figure 2b), we tested if the spreading of γH2AX is regulated by OGT after ionizing radiation. Indeed, our results suggest that diminishing of O-GlcNAcylation by inhibiting OGT displayed significantly larger γH2AX foci at X-ray induced DSBs 24 h after irradiation compared to nontreated cells (Figure 2d), which indicates a further spreading of the phosphorylation around the DSB. In addition, inhibition of OGT led to an increased overall γH2AX foci signal compared to DMSO-treated cells (Figure 2d). As the integrated intensity increased to a similar extent as the nuclear area, this indicates a generally conserved phosphorylation density of H2AX in the chromatin domain.

We further corroborated the relevance of O-GlcNAcylation for the DDR by clonogenic survival assays (Figure 2e), which revealed that a decreased level of O-GlcNAcylation reduced cell survival. The mean inactivation dose of OGT-inhibited cells was 2.04 Gy compared to 2.58 Gy in DMSO-treated control cells. On the other hand, inhibition of OGA displayed decreased radiation sensitivity, indicated by the increased mean inactivation dose of 2.75 Gy after X-ray irradiation. These observations are in agreement with the DSB-rejoining data (Figure 2c).

#### 2.2.2. O-GlcNAcylation Regulates DSB Repair of High-LET Radiation-Induced DSBs

In Figure 1b,c we show that O-GlcNAcylation was specifically located at the DSB sites after iron-ion irradiation, whilst a global increase in O-GlcNAcylation, but no clear localization at DSBs, was observed after X-ray irradiation (Figure 1a). For this reason, we hypothesized that O-GlcNAcylation could be associated with DNA damage complexity and might especially impact on DSB repair after irradiation by charged particles. To test our hypothesis, we examined the kinetics of DSB repair in a cell-cycle-dependent manner using high-LET carbon and helium ion irradiation. The cell-cycle phases were determined using the cell-cycle marker CENP-F.

Modulation of O-GlcNAcylation showed no impact on γH2AX foci formation in HeLa cells after high LET irradiation as seen earlier for X-ray induced DSBs (Figure 3a). Repair kinetics revealed minor effects of modulation of O-GlcNAcylation on the repair of carbon or helium ion-induced DSBs in G1 cells (Figure 3a). However, in S/G2 phase cells inhibition of OGA significantly stimulated the loss of γH2AX foci 24 h after irradiation, whereas OGT inhibition led to increased γH2AX foci numbers at later times post carbon and helium ion irradiation. Interestingly, deficiency of O-GlcNAcylation (OGTi treatment) also displayed higher induction of micronuclei 24 h after both carbon and helium ion irradiation, supporting a diminished repair before going into mitosis (Figure 3b).

### 2.3. O-GlcNAcylation Is Involved in HR

As the impact of O-GlcNAcylation on DSB repair was mainly observed in S/G2 phase, we next addressed if this post-translational modification (PTM) modulates DNA DSB repair via the HR pathway, which is restricted to S and G2 phase due to the requirement of the sister chromatid as a template [30]. To suppress HR, the key factor RAD51 was depleted using siRNA (Figure 4a) and DSB repair was measured by quantifying radiation-induced foci of γH2AX per nucleus after 2 Gy X-ray irradiation with dependence on OGT or OGA inhibition (Figure 4b,c) and the cell-cycle phase. The latter was distinguished with the cell-cycle marker CENP-F.

Modulation of O-GlcNAcylation (OGTi or OGAi), and/or RAD51 depletion, showed no influence on the number of induced DSBs visualized by γH2AX immunostaining 15 min after X-ray irradiation (Figure 4b). As expected, RAD51 depletion did not influence the repair in G1 phase cells (Figure 4c) but showed increased residual damage in S/G2, indicating that part of the repair is based on HR in this fraction of cells (Figure 4c). Similarly, OGT inhibition gave rise to an elevated number of γH2AX foci in S/G2 phase cells 8 h after irradiation, indicating an impairment of DSB repair (Figure 4c). Notably, inhibiting O-GlcNAcylation (OGTi) in RAD51-depleted cells did not further impair DSB repair, suggesting that O-GlcNAcylation was acting in the same pathway as RAD51, i.e., HR (Figure 4c). On the contrary, O-GlcNAc persistence in OGA-inhibited cells supports DSB repair both in mock-depleted as well as in RAD51-depleted cells (Figure 4c). In the RAD51-depleted S/G2-phase cells, the HR defect was almost rescued. Thus, on the one hand data on OGTi and OGTi + RAD51 kd suggest that O-GlcNAcylation is required to allow HR to proceed, yet on the other hand, OGAi and OGAi + RAD51 kd data suggest that constant O-GlcNAcylation allows cells to bypass HR and repair DSBs by different pathways. These complementary results indicate that regulation of HR by O-GlcNAcylation is a sophisticated process that is sensitive to quantity and timing of O-GlcNAcylation.

### 2.4. Inhibiting O-GlcNAc Affects NBS1 Accumulation at DSBs Sites

GlcNAcylation of H2B at S112 is suggested to promote NBS1 recruitment to DSBs [31], a member of the MRN complex. Besides damage-sensing and activation of ATM, this complex participates in the early steps of DSB end resection via the recruitment of activated CtIP, which is a prerequisite for HR repair [8]. In order to test whether O-GlcNAcylation affects NBS1 recruitment to X-ray-induced DSBs in a more direct approach, we performed live-cell imaging experiments using U2OS cells expressing NBS1-2GFP treated with or without OGT inhibitor.

Interestingly, NBS1 protein recruitment (Figure 5a,b) showed similar accumulation kinetics at DSBs during the first 15 min after X-ray irradiation, with a half maximal recruitment time of around 300–400 s independent of OGT inhibition. However, compared to noninhibited cells, diminishing the cellular level of O-GlcNAcylated proteins by inhibiting OGT caused an early loss of NBS1-2GFP signal intensity starting at around 15–20 min after irradiation and revealed a significant deviation (*t*-test) of the two curves at later times (t > 30 min) (Figure 5b). This indicates a more transient binding, or reduced steady state concentration, of NBS1 at DSBs at later times after DSB induction.

### 2.5. CtIP, BRCA1 and MDC1 Are Modified by O-GlcNAcylation in Response to X-ray Irradiation

To gain deeper insight into which DNA damage response or repair proteins are modified by O-GlcNAc in response to ionizing radiation, we performed immunoprecipitation (IP) experiments in HeLa cells. Within immunoprecipitated GlcNAcylated proteins we studied the presence of DSB-repair proteins RPA, CtIP, RAD51 and BRCA1 that are involved in HR and known to be regulated by phosphorylation within the course of HR [32,33,34,35]. The reasoning was that phosphorylation sites are frequently GlcNAcylation sites [16]. We chose MDC1 to serve as a positive control, since it has been shown previously that its GlcNAcylation is increased upon ionizing irradiation [21]. In dependence of X-ray irradiation (10 Gy), we studied the presence of CtIP, BRCA1, RPA, RAD51 and MDC1 in the pool of O-GlcNAcylated proteins 1 h (CtIP, MDC1) or 2 h (BRCA1, RPA, RAD51) after irradiation. The time points were selected according to when the particular protein showed its peak level upon DNA damage [36,37,38,39].

The quality of the IP was ensured by immunoblotting O-GlcNAc with an antibody that was different from the one used for the precipitation (Figure 6a). In agreement with Chen and Yu [21], we detected an increased level of O-GlcNAcylated MDC1 upon irradiation, which demonstrates the validity of our IP approach (Figure 6b). Interestingly, neither RAD51 nor RPA was detected in the O-GlcNAcylated protein fraction (Appendix A), although present in the extract used for the immune-precipitation. This suggests, that in the observed time window, RPA and RAD51 are no major targets for O-GlcNAcylation. Notably, we found that CtIP and BRCA1 became increasingly O-GlcNAcylated after irradiation (Figure 6c,d), which implies that the function of these factors is regulated by this PTM.

### 2.6. O-GlcNAcylation Regulates the Chromatin Status

As O-GlcNAcylation was shown to be a PTM of histone proteins [24], it might interfere with the repair process by influencing the chromatin status and thus, indirectly, the recruitment of repair factors apart from a direct regulation. Hence, we performed fluorescence lifetime microscopy (FLIM) measurements in living HeLa cells treated with OGT inhibitor, OGA inhibitor or DMSO (solvent control) using the fluorescence lifetime of the DNA dye Hoechst 34580 as a chromatin compaction sensor [28].

Typical images obtained from the FLIM measurements of Hoechst 34580 are shown in Figure 7a (intensity, left) and color-coded values of the according fluorescence lifetime are indicated in Figure 7a (right). Whereas the overall staining and the chromatin pattern were similar under the applied conditions, an increased level of O-GlcNAcylated proteins triggered by OGA inhibition caused an elevated fluorescence lifetime (1212 ± 14 ps) in comparison to DMSO control cells (1176 ± 12 ps) (Figure 7b) indicating global chromatin relaxation. In agreement with this observation, we observed a decreasing fluorescence lifetime (1113 ± 16 ps) upon downregulation of O-GlcNAcylation using OGT inhibitor, suggesting the opposite reaction, i.e., a global chromatin compaction. Overall, the chromatin underwent reorganization upon modulating O-GlcNAcylation, proving putative regulatory roles at the chromatin level.

## 3. Discussion

O-GlcNAcylation was first discovered by Torres and Hart [40]. Since then, increasing evidence is accumulating indicating that the PTM O-GlcNAcylation is not only involved in cellular signaling and transcription [16,41] but also plays a role in the DNA-damage response [19,20,21,25,31,42]. Our findings provide further proof that DNA damage induces O-GlcNAcylation after ionizing irradiation, as observed previously after laser microirradiation [21]. Upon ionizing irradiation, we detected an increase of O-GlcNAcylation within nuclei of irradiated cells. Interestingly, this increase was global when damage was induced with sparsely ionizing X-ray irradiation (Figure 1a), but also local at DSBs when DNA damage was induced by densely-ionizing accelerated ions (Figure 1b–d). The increased visibility of O-GlcNAc with heavy-ion-induced and laser-inflicted [21] DNA damage may be due to the locally increased density of lesions compared to X-ray-induced DNA damage, thus causing a high concentration of O-GlcNAc-modified chromatin along the ion trajectory or laser irradiation path. Recruitment of more repair-relevant factors due to the increased DNA-damage density may further add to the improved visibility of O-GlcNAc at charged particle or laser induced DSBs, as we showed that several of these factors are modified by O-GlcNAcylation upon irradiation, e.g., MDC1 [21] as well as CtIP and BRCA1 (Figure 6b–d). The early timepoint (20 min; Figure 1) after which we observed local accumulation (iron ion irradiation) or global accumulation (X-ray and-iron ion irradiation) suggests that this post-translational modification is directly involved in the early steps of break processing, e.g., via activation or recruitment of repair factors.

Despite a direct regulation of protein activity via O-GlcNAcylation, potential crosstalk between O-GlcNAcylation and phosphorylation on the amino acids Ser and Thr at the same, adjacent or distant sites was proposed to play a regulatory role [19]. As many repair factors are known to be modified by phosphorylation events at regulatory sites, this crosstalk mechanism might be important in the DNA-damage response. Along this line, Chen and Yu previously revealed that OGT directly localizes to DNA damage sites and dynamically modifies the DSB response factors H2AX and MDC1 in competition with phosphorylation [21]. This is in agreement with our observation that OGT inhibition caused significantly bigger and intensive γH2AX foci 24 h after irradiation in S/G2 phase cells (Figure 2d). The similarity of staining density despite the increased area points to a larger spreading of the γH2AX signal into the surrounding chromatin rather than an increased local chromatin decompaction.

To understand the consequences of radiation-induced O-GlcNAcylation in more detail, we investigated how manipulating O-GlcNAcylation affects DSB repair capacity in a cell-cycle dependent manner. We found that O-GlcNAcylation impacts on DSB repair. In general, similar results were obtained after low (Figure 2b–c) and high LET (Figure 3a) irradiation, indicating that, besides the observed differences in radiation induced O-GlcNAcylation patterns, the modulation is not strictly dependent on radiation quality. Of note, a significant repair modulation by O-GlcNAcylation was observable only in S/G2 phase cells. In S/G2 phase cells, O-GlcNAc transferase inhibition (OGTi) led to a diminished DSB repair after X-ray and charged particle irradiation by almost two-fold compared to solvent treated cells (DMSO) (X-rays: 1.7x; carbon/helium ions: 1.8x). On the other hand, O-GlcNAcase inhibition (OGAi) promoted repair of X-ray and charged particle-induced DSBs in S/G2 cells by about two-fold compared to solvent-treated cells (DMSO) (X-rays: 2.6x; carbon/helium ions: 2.3x). The influence on DSB repair in S/G2 cells implies that O-GlcNAcylation may regulate a subset of DSB repair processes specific to these cell-cycle phases, i.e., HR because it needs replicated DNA as a template. The finding that modulating O-GlcNAcylation after X-rays shows no significant impact on DSB repair in G1-phase cells, argues against a major regulation of NHEJ as described by Wang et al. [31]. However, even if only a limited proportion of repair in S/G2 is significantly affected by O-GlcNAcylation in an asynchronous population, its relevance is demonstrated by an increased radiosensitivity after OGT inhibition measured by clonogenic survival (Figure 2e). In addition to DNA repair, we further have to consider that radiation-dependent O-GlcNAcylation may influence other processes, e.g., transcription [23,24].

The MRN (MRE11/Rad50/NBS1) complex factor NBS1 has been proposed as a promoter of resection at DSBs via its interaction with CtIP [43], which is a prerequisite for HR [44]. Our live-cell study revealed recruitment kinetics for NBS1 with a half maximal recruitment around 300–400 s, which is in good agreement with the recruitment time measured previously for X-rays [7], but somewhat slower than for heavy ion irradiation, which shows an LET-dependent acceleration of NBS1 foci formation [6]. Interestingly, the initial recruitment kinetics of NBS1 was unaltered upon OGT inhibition, pointing to no disturbance in damage recognition itself. However, a reduced retention of NBS1 at radiation-induced DSBs became obvious when OGT was inhibited (Figure 5), which is in line with the observed impact of the O-GlcNAcylated histone H2B on NBS1 interaction and accumulation at DSBs sites [31]. Thus, diminished retention of NBS1 might impede, especially, resection-dependent repair such as HR, a pathway that is used in our cell system and is stimulated by O-GlcNAcylation (Figure 4c, right). This indicates that O-GlcNAcylation by OGT supports HR. However, increased O-GlcNAcylation by OGA inhibition still showed a stimulation in DSB repair even under RAD51 knockdown conditions. This indicates that regulation of HR by O-GlcNAcylation is not a simple on-off process but a sophisticated regulation tool that affects several proteins. Furthermore, these data point to an additional alternative repair mechanism regulated by O-GlcNAcylation in S/G2. However, we have to acknowledge that the application of the inhibitors well before irradiation might have already changed the pre-irradiation conditions, and not only modulate the direct radiation response.

Regulation of HR by O-GlcNAcylation is also corroborated by our immunoprecipitation data, demonstrating that besides MDC1, which was known to be O-GlcNAcylated in a DNA-damage dependent manner, also CtIP and BRCA1, two proteins involved in HR relevant processes, are modified by O-GlcNAcylation in a radiation-dependent manner. CtIP is required for the initiation of resection, a necessary precursor for homology search and D-loop formation in HR in G2 phase [44]. Interestingly, CtIP’s resection-relevant activity is regulated by phosphorylation at specific sites, which controls its interaction with NBS1 and BRCA1, respectively [33,45,46,47]. Similar to CtIP, BRCA1′s activity is regulated via phosphorylation at different sites [48]. Therefore, it is conceivable that, similar to MDC1 and H2AX [21], CtIP’s and BRCA1’s O-GlcNAcylation may serve to regulate and fine tune their phosphorylation guided activity.

As a putative level of DDR regulation by O-GlcNAcylation in addition to the direct modulation of repair factors, and in line with the observation of histone proteins being targets for O-GlcNAcylation [24], we also found that chromatin compaction is modulated by OGT and OGA activity. We showed that inhibiting OGT leads to more condensed chromatin, whilst inhibition of OGA induces a global chromatin relaxation. As it was found that, upon DNA damage, chromatin decompaction is stimulated globally [28,49], but also locally [27,50,51], and since it is known that within repair of DNA damage the chromatin status is crucial for the repair-pathway choice and the recruitment of DSB repair factors [26,52], we conclude that O-GlcNAcylation may also impact on the DSBs response by regulating chromatin remodeling. This is supported by several findings that OGT and OGA modulate recruitment, stability and activity of key chromatin regulators [23,24]. Taken together, the sophisticated regulation of HR by O-GlcNAcylation most likely comprises modifying directly-acting DDR factors, but also chromatin factors and chromatin components that facilitate and control HR by establishing optimal structural conditions.

Whereas molecular details of repair regulation via the PTM O-GlcNAcylation still require further investigation, our findings demonstrate that O-GlcNAcylation modulates repair of ionizing radiation-inflicted DSBs in multiple ways by orchestrating the localization of repair relevant factors at DNA lesions, by influencing the repair and repair-pathway choice in a cell-cycle dependent manner, by influencing radiosensitivity and by modifying chromatin to favor DNA repair. Thus, it contributes in multiple ways to maintaining genome stability.

## 4. Materials and Methods

### 4.1. Cell Culture, Inhibitor Treatment and siRNA Transfection

The human cervix epitheloid carcinoma cell line HeLa CCL-2 (ATCC, Wesel, Germany), human breast cancer cell line MCF-7 (Leibniz Institute DSMZ, Braunschweig, Germany) and human osteosarcoma cell line U2OS NBS1-2GFP (kindly provided by Claudia Lukas, Faculty of Health and Medical Sciences, University of Copenhagen, Copenhagen, Denmark), which stably expresses NBS1-2GFP, were maintained in 4.5 g/L D-Glucose DMEM medium (Gibco, Carlsbad, CA, USA) supplemented with 10% fetal calf serum (FCS) (Biochrom, Berlin, Germany) or 20% FCS (MCF-7) in a humidified incubator with 5% CO_2_ at 37 °C. Cells were treated with 12.5 µM of O-GlcNAc transferase (OGT) inhibitor ST060266 (TimTec, Newark, DE, USA) or 100 µM of O-GlcNAcase (OGA) inhibitor PUGNAc (Sigma-Aldrich, Munich, Germany) for 24 h prior to irradiation, and maintained up to 24 h after irradiation. For RAD51 knockdown, HeLa cells were transfected with RAD51 siRNA (final concentration 50 nM; GAGCUUUGACAAACUACUUCdTdT) (Eurofin Genomics, Ebersberg, Germany) 48 h before irradiation. The knockdown was verified by western blot analysis.

### 4.2. Irradiation

Cells were irradiated with X-rays (X-ray tube MXR 320-26, Seifert/GE, Germany) with the stated doses at a voltage of 250 kV and a current of 16 mA. Irradiation of cells with accelerated ions was carried out at the UNILAC linear accelerator of the GSI Helmholtz Center for Heavy ion Research (Darmstadt, Germany). Cells were irradiated perpendicularly with different charged particles: carbon (primary energy 11.4 MeV/nucleon, 5 × 10^6^ p./cm^2^, 168 keV/µm), iron (primary energy 11.4 MeV/nucleon, 5 × 10^6^ p./cm^2^, 2875 keV/µm), or helium (primary energy 3.6 MeV/nucleon, 5 × 10^6^ p./cm^2^, 82 keV/µm).

### 4.3. Immunoprecipitation and Western Blotting

HeLa cells (10^6^) were seeded in 10 cm petri dishes two days before the experiment. Protein was extracted at indicated time points after 10 Gy of X-rays with RIPA lysis buffer (20 mM Tris-HCl, 150 mM NaCl, 1% NP-40, 0.5% Na-deoxycholate (*w*/*v*), 0.1% SDS, 1 mM EDTA, 50 µM PUGNAc, protease and phosphatase inhibitors) on ice. Protein (1 or 2 mg) was immunoprecipitated with 10–20 µg of O-GlcNAc RL2 antibody (MA1072, Invitrogen/Thermo Fisher Scientific, Waltham, MA, USA) or an equal amount of IgG1 antibody (Mouse IgG1 Isotope Control, MA110407, Invitrogen/Thermo Fisher Scientific, Waltham, MA, USA) coupled with Protein G (Dynabeads Protein G Immunoprecipitation Kit, Invitrogen/Thermo Fisher Scientific, Waltham, MA, USA) for 1 h at RT. Following washing steps, provided by the kit, proteins were mixed with 3× blue loading buffer (Cell Signaling Technology, Frankfurt, Germany). The target antigen was eluted by boiling the samples (containing the loading buffer) for 15 min. Magnetic beads were separated from samples using a magnetic rack. Proteins were separated in miniprotean TGX Precast polyacrylamide gels (Bio-Rad, Munich, Germany) (4–6% or 10%). In the case of whole cell extracts, 20 µg protein were loaded per lane. Proteins were efficiently transferred to a nitrocellulose membrane using a transblot turbo RTA transfer kit (Bio-Rad, Munich, Germany). The membrane was blocked with 5% low fat milk or 5% BSA in TBS-T for 1 h at RT. Antibodies were diluted in TBS-T, and primary antibody incubation was performed overnight at 4 °C. The primary antibodies were: αRAD51 (rabbit, ab133534, Abcam, Cambridge, UK) at dilution 1:5000, αRPA 32 kDa subunit (9H8) (mouse, sc-56770, Santa Cruz, Heidelberg, Germany) at dilution 1:200, αBRCA1 (mouse, sc-6954, Santa Cruz, Heidelberg, Germany) at dilution 1:200, αCtIP (mouse, sc-271339, Santa Cruz, Heidelberg, Germany) at dilution 1:200, αMDC1 (rabbit, PA5-97022, Invitrogen/Thermo Fisher Scientific, Waltham, MA, USA) at dilution 1:1000, αO-GlcNAc (CTD110.6, mouse, sc-59623, Santa Cruz, Heidelberg, Germany) at dilution 1:500. Secondary antibodies were: HRP goat anti mouse or rabbit (LI-COR, Bad Homburg, Germany) at dilution 1:10000. Western blots were developed with ECL reagents (Roche, Mannheim, Germany).

### 4.4. Immunofluorescence Staining, Microscopy and Data Analysis

For the iron ion-induced colocalization experiment, the soluble cytoskeleton proteins were extracted with a cytoskeleton buffer and a cytoskeleton stripping buffer. Then, thecells were fixed with STRECK fixation buffer according to Jakob et al. [53]. In all the other experiments, cells were fixed with 2% paraformaldehyde for 15 min, permeabilized with 0.1% Triton X-100 for 10 min and blocked with 0.4% BSA in PBS at least 20 min. All primary and secondary antibodies were diluted in 0.4% BSA in PBS. Primary antibodies were: αγH2AX clone JBW301 (Ser139, mouse, 05-636, Millipore, Darmstadt, Germany) at 1:500 dilution, αCENP-F (rabbit, NB500-101, Novus Biologicals, Centennial, CO, USA) at 1:750 dilution, αO-GlcNAc RL2 (mouse, sc-59624, Santa Cruz, Heidelberg, Germany) at 1:200 dilution, αO-GlcNAc CTD110.6 (mouse, sc-59623, Santa Cruz, Heidelberg, Germany) at 1:500 dilution, α53BP1 (rabbit, ab 36823, Abcam, Cambridge, UK). Secondary antibodies were: Alexa 488-conjugated goat αmouse (Invitrogen/Thermo Fisher Scientific, Waltham, MA, USA) at dilution 1:400, Alexa 568-conjugated donkey αrabbit (Life Technologies, Darmstadt, Germany) at dilution 1:400. DNA was counterstained with DAPI (AppliChem, Darmstadt, Germany) at a concentration of 1 µg/mL.

### 4.5. Scoring of Micronuclei (MN)

Micronuclei were visualizing by DAPI staining and scored in 150–250 nuclei. The criteria for MN were the following. The diameter of MN was less than 1/3 of the “main” nucleus. The color of MN was the same as, or lighter than, the “main” nucleus. The position of MN was close to the “main” nucleus, i.e., the distance to the “main” nucleus was smaller than the radius of the “main” nucleus. A “main” nucleus may have more than one MN.

### 4.6. Quantifying DSB Rejoining

To study repair of radiation-induced DSBs, we used the γH2AX foci assay [9], in which the DSB marker γH2AX was visualized by immunofluorescence staining and enumerated per nucleus. Data were corrected by subtracting the number of background foci of nonirradiated cells.

### 4.7. Clonogenic Survival Experiment

Clonogenic survival was determined according to Wang et al. [54]. HeLa cells (250,000) were seeded in 25 cm^2^ culture flasks and allowed to adhere for 24 h. Then, cells were treated or nontreated with OGT or OGA inhibitors for 24 h and subsequently irradiated with 0, 1, 2, 3, 4 or 6 Gy of X-rays. Irradiated cells were trypsinized and reseeded in triplicate in medium containing DMSO, OGA or OGT inhibitor to obtain 100 colonies. The cell number to be seeded was determined with respect to the plating efficiency and dose. The cells remained in culture for 10 days. Inhibitors were not refreshed during this time. After 10 days of incubation, cells were stained with methylene blue and the colonies containing at least 50 cells were counted to determine the survival rate. The mean inactivation dose was calculated according to Fertil et al. [55].

### 4.8. Live Cell and FLIM Experiment

For live-cell microscopy and for FLIM, U2OS NBS1-2GFP and HeLa cells, respectively, were seeded in 35 mm glass-bottom Petri dishes two days before the experiment (1.2 × 10^5^ cells). The inhibitor treatment was performed as described in 4.7. To monitor NBS1 recruitment, live cell imaging in combination with X-ray irradiation was done using the equipment according to Jakob et al. [7]. AndorIQ software (version 1.10) was used to acquire live cell images up to 45 min after 1 Gy of X-ray irradiation. To perform FLIM, living HeLa cells were stained with 1 µM Hoechst 34580 (Biomol GmbH, Hamburg, Germany) for 1 h and the culture medium was refreshed after the incubation period. The experiment was carried out using single photon counting (TCSPC, a DCS 120 scan head; Becker & Hickl, Berlin, Germany) according to Abdollahi et al. [28]. A stage climate chamber (Tokai Hit, Fujinomiya-shi, Shizuoka, Japan) was used to keep the temperature (37 °C), humidity and CO_2_ (5%) stable. Laser power was set to yield 10^5^–10^6^ photons/s. The collection time was 30 s in FIFO mode.

### 4.9. Image Analysis

Data were plotted using GraphPad Prism software (Version 8, San Diego, CA, USA) or Excel (Version 2016, Microsoft Corporation, Redmond, WA, USA). For image analysis, Image J 1.52a [56] including the StackReg plugin (Philippe Thévenaz, Lausanne, Switzerland) was used. For the FLIM experiment, image analysis was done with SPCImage (Version 6.4, Becker & Hickl, Berlin, Germany).

### 4.10. Statistical Analysis

To calculate the average, standard deviation (SD), or standard error of the mean (SEM) of two independent experiments, we used the following equations:

${\bar{x}}_{1,2}$ is the combined mean of experiments 1 and 2, ${\bar{x}}_{1}$ the mean of data in experiment 1, ${\bar{x}}_{2}$ the mean of data in experiment 2, *N*_1_ the sample size in experiment 1, *N*_2_ the sample size in experiment 2, *s*_1,2_ the combined SD of experiment 1 and 2, *s*_1_, SD of experiment 1, *s*_2_ the SD of experiment 2, and *s*?_x_ the standard error of the combined mean.
$$x_{1,2}=\frac{N_{1}\times {\bar{x}}_{1}+N_{2}\times {\bar{x}}_{2}}{N_{1}+N_{2}}$$
$$s_{1,2}=\sqrt{\frac{N_{1}\times (s_{1}^{2}+{\left({\bar{x}}_{1}-{\bar{x}}_{1,2}\right)}_{}^{2}+N_{2}\times (s_{2}^{2}+{\left({\bar{x}}_{2}-{\bar{x}}_{1,2}\right)}_{}^{2}}{N_{1}+N_{2}}}$$
$$s_{\bar{x}}=\frac{s_{1,2}}{\sqrt{N_{1}+N_{2}}}$$

To calculate the mean survival fraction and SD of two independent survival experiments, all data were pooled and the mean and SD calculated.

For *t*-test analyses, GraphPad Prism 8 (GraphPad Software) or Excel 2016 (Microsoft Office) was used.

## Figures and Tables

**Figure 1 ijms-22-05715-f001:**
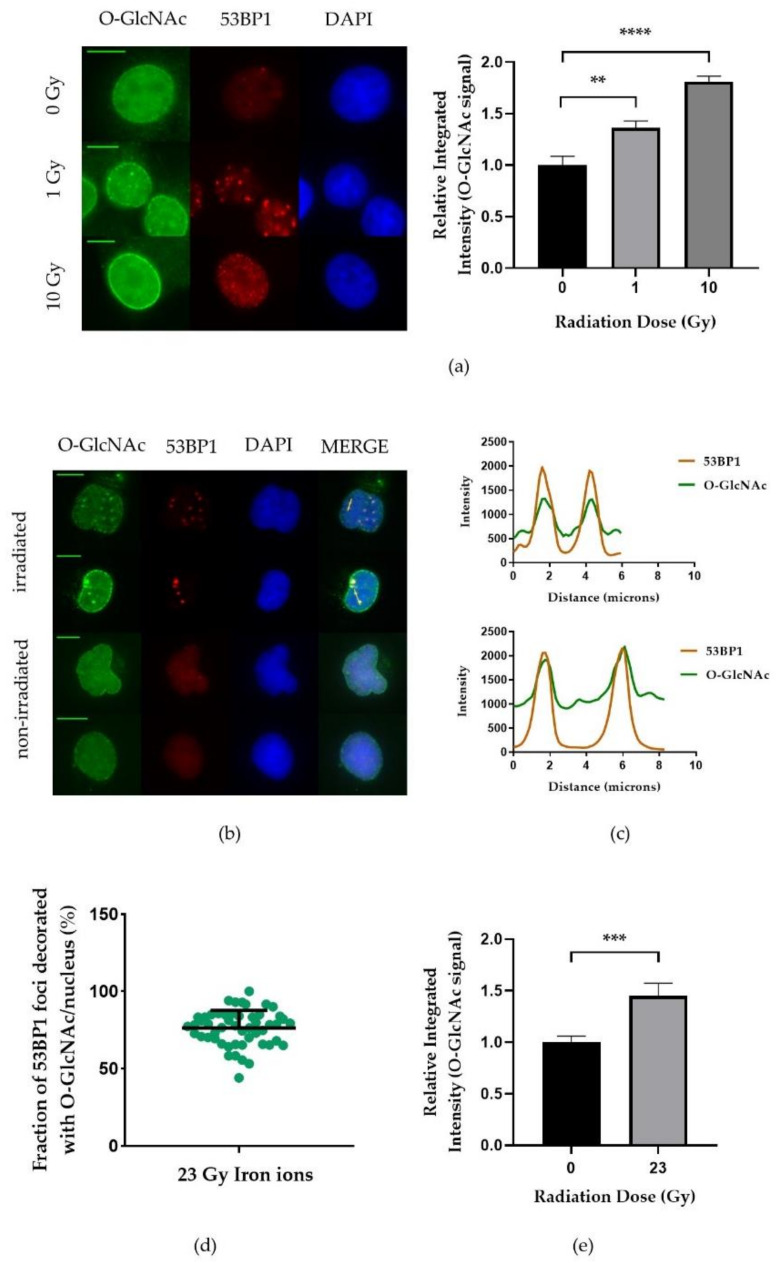
Nuclear O-GlcNAc level increases after X-ray and heavy-ion irradiation, and specifically locates at DSBs after heavy-ion irradiation. Cells were irradiated with X-rays (**a**) or iron ions (**b**–**e**). O-GlcNAc (green) and the DSB marker 53BP1 (red) were visualized by immunofluorescence staining. DNA was counterstained with DAPI. (**a**) Immunofluorescence stained MCF-7 cells (**left**; scale bar 10 µm) were used to measure the integrated fluorescence intensity of X-ray induced O-GlcNAcylation (**right**). For each condition, data represent mean ± SEM of two independent experiments with 25 nuclei each. (**b**) Exemplary immunofluorescence images showing O-GlcNAc and 53BP1 in irradiated and nonirradiated HeLa cells. (**c**) Intensity profiles of O-GlcNAc and 53BP1 signals within HeLa cells (yellow lines within immunofluorescence images in (**b**). (**d**) Quantification of O-GlcNAc decorated DSBs, which were visualized by 53BP1 immunofluorescence staining in irradiated HeLa cells (mean ± SD, *n* = 50 nuclei). (**e**) Quantitative analysis of the integrated fluorescence intensity of the O-GlcNAc signal in iron-ion irradiated HeLa cells. For each condition, data represent mean ± SEM of two independent experiments with 25 nuclei each. *p* values are based on *t*-test analyses, asterisks indicate ** *p* < 0.01, *** *p* < 0.001, and **** *p* < 0.0001.

**Figure 2 ijms-22-05715-f002:**
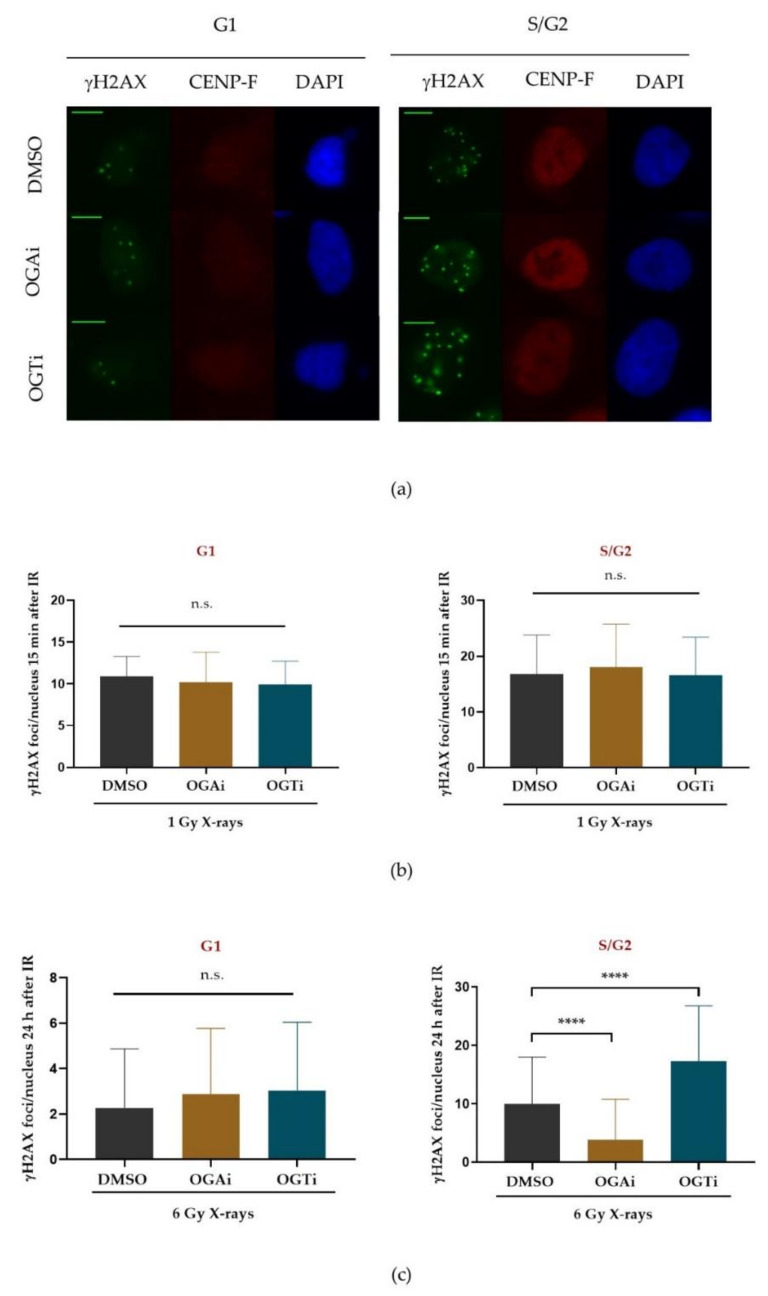
O-GlcNAcylation impacts on DNA DSB repair and cell survival after X-ray irradiation. (**a**) Representative images of the DSB marker γH2AX in O-GlcNAc transferase inhibitor (OGTi) or O-GlcNAcase inhibitor (OGAi)-treated S/G2 phase or G1 phase Hela cells 24 h post irradiation (6 Gy). The cell-cycle phases were distinguished by immunofluorescence-staining the cell-cycle marker CENP-F and counterstaining DNA with DAPI. CENP-F positive but not M phase: S and G2 cells; CENP-F negative: G1 cells. Scale bar: 10 µm (**b**) Quantification of radiation-induced γH2AX foci per nucleus 15 min after 1 Gy X-ray irradiation (mean ± SD of two independent experiments. In each experiment 50 nuclei per condition were analyzed). (**c**) Quantification of γH2AX foci per nucleus 24 h after 6 Gy X-ray irradiation (as in (**a**)). For each condition, data represent mean ± SD of two independent experiments with 50 nuclei each. (**d**) γH2AX foci size and integrated fluorescence intensity of γH2AX foci in S/G2 cells. For each condition, data represent mean ± SEM of two independent experiments with 30 nuclei each. (**e**) Clonogenic survival of OGTi, OGAi, or nontreated (DMSO) HeLa cells irradiated with X-rays (mean ± SD of two independent experiments performed in triplicate). (**b**–**e**) *p* values are based on *t*-test analyses, asterisks indicate * *p* < 0.05, ** *p* < 0.01, and **** *p* < 0.0001; n.s.: not significant.

**Figure 3 ijms-22-05715-f003:**
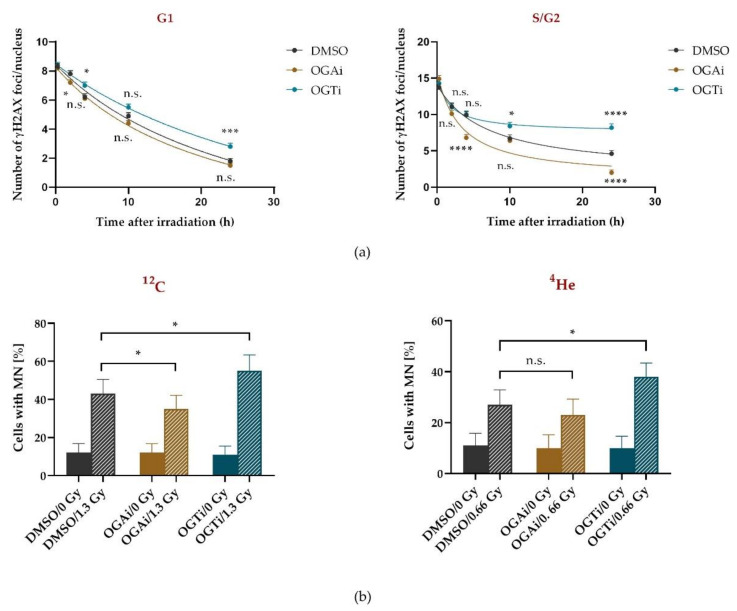
Loss of O-GlcNAcylation led to impaired DSB repair in S/G2 phase after heavy-ion irradiation. HeLa cells treated with an O-GlcNAc transferase inhibitor (OGTi) or O-GlcNAcase inhibitor (OGAi), were exposed to carbon ion or helium ion irradiation (fluence 5 × 10^6^ p/cm^2^) and fixed at the indicated time points. (**a**) Quantification of γH2AX foci per nucleus after carbon ion or helium ion irradiation in S/G2 or G1 phase cells. The cell-cycle phases were determined as in Figure 2 using the cell-cycle marker CENP-F (mean ± SEM; 50 nuclei per condition and experiment, results of one helium and one carbon-ion experiment are averaged) (**b**) Fraction of micronuclei (M.N.) 24 h after heavy-ion irradiation. Mean ±binomial error; at least 150 cells per condition of a single experiment. (**a**,**b**) *p* values are based on *t*-test analyses, asterisks indicate * *p* < 0.05, *** *p* < 0.001 and **** *p* < 0.0001; n.s.: not significant.

**Figure 4 ijms-22-05715-f004:**
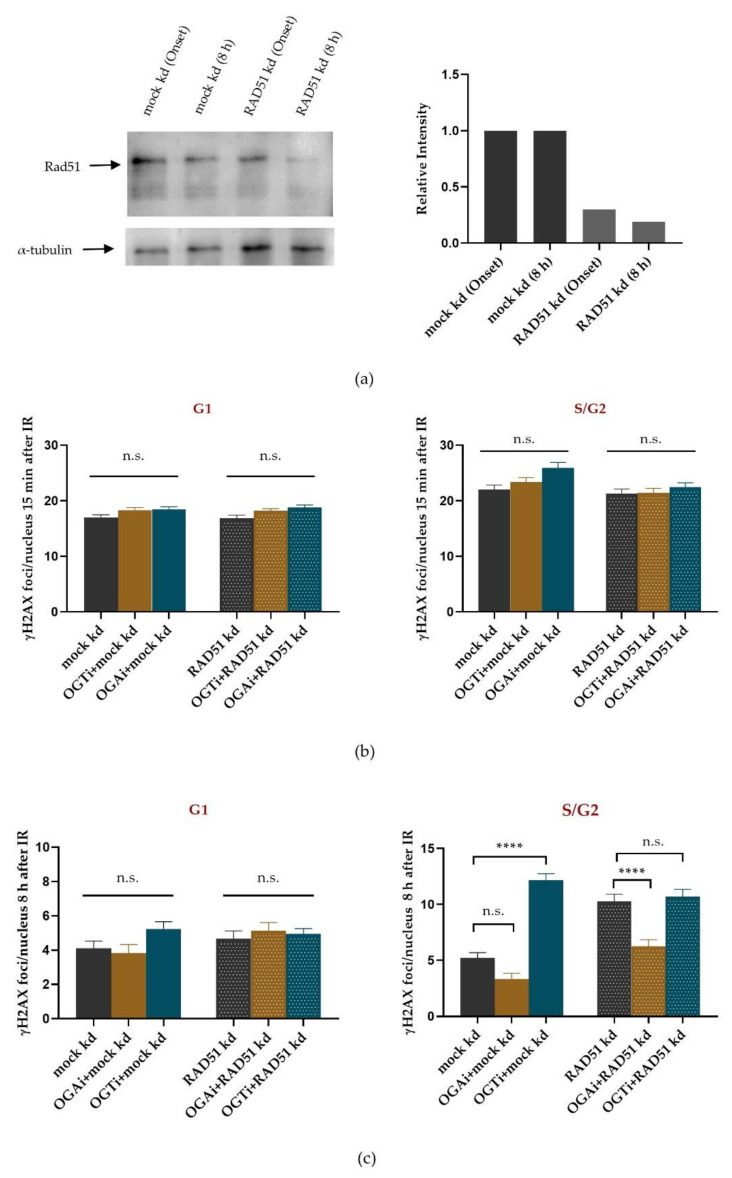
DSB repair via HR is modulated by O-GlcNAcylation. HeLa cells Rad51-depleted or mock-depleted by RNAi (RAD51 kd, mock kd) and treated with or without OGA or OGT inhibitor (OGAi, OGTi) were exposed to 2 Gy X-ray irradiation and fixed at the indicated time points. (**a**) Western blot analysis of RAD51 knockdown efficiency and its quantification are shown. (**b**,**c**) Quantification of γH2AX foci per nucleus in G1 and S/G2 cells 15 min (**b**) and 8 h (**c**) after X-ray irradiation, respectively. The cell-cycle phases were determined as in Figure 2, using the cell-cycle marker CENP-F. For each condition, data represents mean ± SEM of two independent experiments with 50 nuclei each. (**b**,**c**) *p* values are based on *t*-test analyses, asterisks indicate **** *p* < 0.0001; n.s.: not significant.

**Figure 5 ijms-22-05715-f005:**
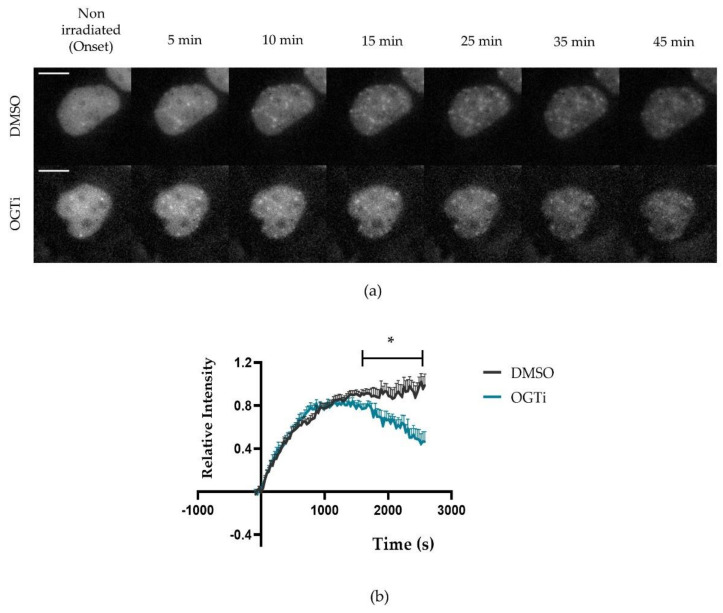
Effect of OGT inhibition on NBS1 recruitment after X-ray irradiation. With live-cell microscopy of U2OS cells expressing NBS1-2GFP the recruitment kinetics and binding of NBS1-2GFP to DSBs were detected within 45 min after 1 Gy X-ray irradiation in DMSO (control) or OGT inhibitor (OGTi)-treated cells. (**a**) Typical images of NBS1-2GFP signal at different time points after irradiation in OGTi treated and nontreated cells. Scale bar: 10 µM. (**b**) Relative NBS1-2GFP accumulation to radiation induced DSBs in control and OGTi-treated cells. The graphs show mean NBS1-2GFP intensity ±SEM of two independent experiments. In each experiment, 20 nuclei were analyzed per condition. Statistical analysis by *t*-test: >30 min, * *p* < 0.05.

**Figure 6 ijms-22-05715-f006:**
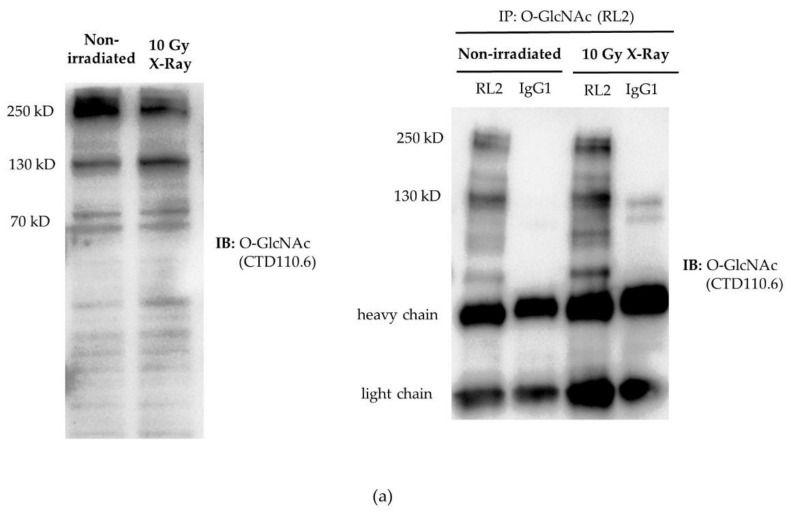
CtIP and BRCA1 are O-GlcNAcylation targets in dependence of irradiation. Immunoprecipitation (IP) of GlcNAcylated proteins was carried out in extracts of irradiated or nonirradiated HeLa cells 1 h after 10 Gy X-ray irradiation using O-GlcNAc-specific antibody (RL2); as a control, IP was performed with nonspecific IgG1 antibody. (**a**) Immunoblotting (IB) of whole cell extracts verified that the extracts contained O-GlcNAcylated proteins (left). Immunoblotting of the O-GlcNAc IP validated that the RL2 antibody specifically precipitated O-GlcNAcylated proteins (right). O-GlcNAc specific antibody: CTD110.6 (**b**) Immunoblot to detect MDC1 in immunoprecipitated O-GlcNAc-modified proteins and quantification of the MDC1 signal, (**c**) as in (**b**) but detection of CtIP, (**d**) as in (**b**) but cells were harvested 2 h after irradiation and BRCA1 was detected. The asterisk (*) indicates the BRCA1 signal. Data represents mean ± SEM of two independent experiments. The signal intensity of irradiated samples was normalized to the nonirradiated signal.

**Figure 7 ijms-22-05715-f007:**
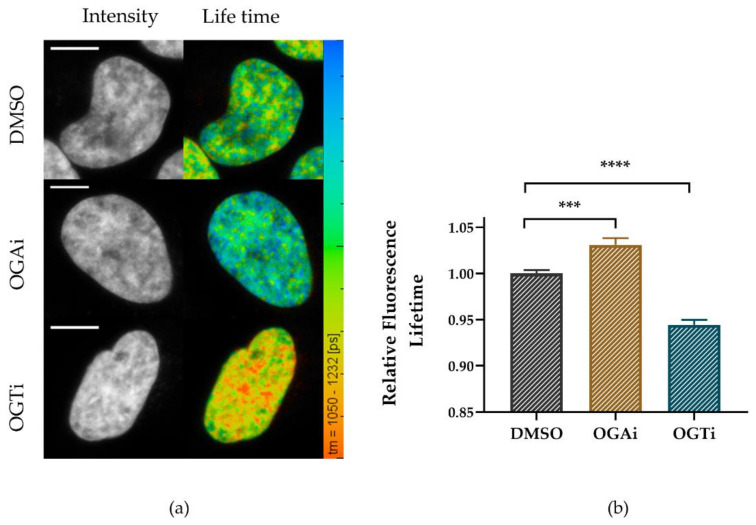
Changing the level of O-GlcNAcylation modulates chromatin compaction. Living HeLa cells were treated or no-treated with OGT inhibitor (OGTi) or OGA inhibitor (OGAi) for 24 h. DNA was stained with Hoechst 34580. (**a**) Representative photon (left) and fluorescence lifetime images on a color-coded scale (1050–1232 ps), (right). Scale bar: 10 µm. (**b**) Quantification of relative Hoechst 34580 fluorescence lifetime in O-GlcNAc-manipulated or control cells. The absolute value of fluorescence lifetime was normalized to control values. For each condition, data represent mean ± SEM of two independent experiments with 25 nuclei each. *p* values are based on *t*-test analyses; asterisks indicate *** *p* < 0.001 and **** *p* < 0.0001.

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
