# Peer review of "O-GlcNAcylation Affects the Pathway Choice of DNA Double-Strand Break Repair"

_ijms, 2021, doi:10.3390/ijms22115715_

Round 1

Reviewer 1 Report

Averbeck et al. have investigated the role of  O-linked β-N-acetylglucosaminylation in the DNA damage response to low and high LET irradiation. They provide evidence that O-GlcNAcylation is important for the progression of HR DNA repair in late S/G2 phase of the cell cycle and show that this posttranslational modification affects the fidelity CtIP and BRCA1 HR repair factors and also influences the progression of HR through modulation of chromatin organization. It is convincingly shown that O-GlcNAcylation contributes in multiple ways to genome stability.  In all, this is an excellent paper that fits the scope of the journal. Before it can be accepted for publication a few points should be addressed.

Line (L) 35: “… genome instability, cellular senescence and apoptosis, which can ultimately lead to cancer” - The authors should rephrase this sentence as apoptosis is a mechanism that does not lead to cancer but counteracts cancer development.

L 44-47: Insert a reference that supports the facts/conclusions put forward in this sentence.

L56: “…the (later) S…“ – change to ‘(late) S phase’

L 106: „iron ions irradiation“ => ‘iron-ion irradiation’

L 127 2.2.1.: in this section we learn that calls in different cell cycle stages were analyzed. But the ‘how’ is missing. L144: Please add a sentence that explains how the cells in the different cell cycle phases were identified.

L156-158: Sentence is too long and hard to understand. Please reword.

L212, Fig.2b, c, statistics:  Please explain how the SEM shown for 2 independent experiments was derived. SEM based on 3 biological repeat experiments would be better. Or are the error bars shown based on the combined data of all nuclei of the 2 experiments? If so, give SD. Please explain; possibly by a statistics part that may be added to Mat&Methods.

Fig.2e, L 218:  “…(mean ± SEM of 2 independent experiments performed in triplicates)”. Again, how was the SEM derived from 2 experiments? If possible, it would be more sound to give the SD of the 6 data points (out of 2 experiments). Please correct/explain.

L 258: Mention here how the S/G2 or G1 phase cells were identified

L 264: Write out PTM, as it is cryptic here.

L 278: “OGA inhibition, which prevents O-GlcNAc removal, supports DSB repair both in …” Not easy to understand for the non-initiated reader. How about ‘… O-GlcNAc persistence in OGA inhibited cells …’?

L462, Fig. 7: Data and SEM from 25 nuclei without biological repeat experiments? It would be better to show SD as error bars. The authors are strongly encouraged to add additional data from independent repeat experiments were added, which would strengthen this analysis.

Reviewer 2 Report

The manuscript “O-GlcNAcylation affects the pathway choice of DNA double-2 strand break repair” written by S. Averbek et al. brigs interesting findings on multiple roles of O-GlcNAcylation in repair of radiation induced DNA double strand breaks (DSBs). While the participation of O-GlcNAcylation in DSB repair has already been reported after laser microirradiation, in the present manuscript, the authors significantly develop the knowledge using sparsely ionizing g-rays and densely ionizing (high-LET) ions, together with OGT and OGA inhibitors. In a nicely designed study, the authors demonstrate involvement of O-GlcNAcylation in regulation of homologous recombination (HR) and its potential shift to another (novel?) repair mechanism. The obtained results are novel, scientifically sound, and together well support the postulated conclusions. The manuscript is clearly and nicely written. Considering these facts, I recommend the manuscript for publication in IJMS. Nevertheless, I have few minor issues to be fixed before publication.

  1. Pg. 2, ln 56 (and then in the Discussion): HR is presented as an error-free process, which only proceeds in S/G2. While this is true in general, active genes can repair DSBs by HR also in G1 (using nascent RNA as a template) and HR could also be error prone, if problems in chromatin domain reorganization appear during repair (discussed, e.g., in Falk, Hausmann, Cancers, 2021). As the present manuscript also deals with the relevance of chromatin architecture changes in HR regulation, perhaps, the situation could be explained by 1 or 2 more sentences.
  2. Fig 1 – Is it possible to include the same graph as Fig. 1d also for g-rays? The authors suggest that O-GlcNAcylation could be detected at DSB (53BP1) sites only upon high-LET irradiation, which they attribute to higher DSB concentration along the particle track. As g-rays may occasionally also induce (simple) DSB clusters, it might be interesting to see if at least some 53BP1 foci colocalize with O-GlcNAcylation upon g-irradiation. If no such foci exist, mentioning this fact in the text is sufficient.
  3. Fig 1e: Adding a graph title or describing the horizontal axis, e.g., as “Relative Integrated Density, O-GlcNAcylation” would improve orientation in the Figure 1 without searching for the description in the legend.
  4. Pg 5 ln 139: “In the first step”
  5. Fig 2c: For the control cells (DMSO-treated), there are about 2 DSBs per nucleus on average at G1 phase of the cell cycle; this number in S/G2 is about 10 for the same dose. This observation is surprising concerning 2 times higher (only) amount of DNA. This may point to a replication stress in S-phase. If so, a few nuclei, which proportions may just by chance significantly differ between experiments/treatments, may dramatically influence the average number of foci/DSB per nucleus. If this is the true scenario, it should be stated in the text (or demonstrated on a figure panel) that the proportions of nuclei with high DSB numbers are comparable for different treatments (to exclude the possibility that the differences observed for DMSO, OGAi and OGTi is due to different numbers of cells with replication stress involved in the analyzed cell populations. If there is another reason for this high difference between G1 and S/G2 cells, it should be briefly explained.
  6. Figure 2d: could results for OGAi also be displayed?
  7. p7, ln 223: “but no clear correlation was observed” – explain please the exact meaning
  8. Fig 3B: explain please how the fraction of MN is defined. About 10% for non-irradiated controls seems to me quite high (but not impossible) –just to be sure how the numbers were counted.
  9. p 15, ln 494: the authors probably mean “OGT inhibition caused” instead of “OGT caused”
  10. p 16, ln 561 – The influence of chromatin architecture on DSB repair pathway selection has been recently in detail discussed in Falk, Hausmann, Cancers 2021. Also considering what is stated later in the discussion of the present manuscript (ln 565 – end), an interesting idea emerges that O-GlcNAcylation may influence chromatin architecture at the site of DSB, and, in turn, the architecture of IRIF foci, which favors binding of HR-specific proteins. At the same time, O-GlcNAcylation may “authorize” some of these HR-specific proteins (CtIP, BRCA1) for binding to IRIFs. Hence, O-GlcNAcylation may unify in a synergic way biochemical and physical (structural) regulation of DSB repair selection. Perhaps the authors would like to add some brief discussion on this issue.
  11. p21, ln 638-639: OGTi and OGTi were just left in the medium during the experiment or were the inhibitors added repeatedly during the experiment duration.
  12. Maybe I just skipped this information but I cannot find how G1 vs S/G2 cells were identified/selected?
